# Modelling Cancer Pathophysiology: Mechanisms and Changes in the Extracellular Matrix During Cancer Initiation and Early Tumour Growth

**DOI:** 10.3390/cancers17101675

**Published:** 2025-05-15

**Authors:** Luis Larrea Murillo, Megan Green, Niall Mahon, Alberto Saiani, Olga Tsigkou

**Affiliations:** 1Department of Materials, School of Natural Sciences, Faculty of Science and Engineering, The University of Manchester, Manchester M13 9PL, UK; megan.green@manchester.ac.uk (M.G.);; 2The Henry Royce Institute, Royce Hub Building, Manchester M13 9PL, UK; 3Manchester Institute of Biotechnology (MIB), The University of Manchester, Manchester M1 7DN, UK

**Keywords:** cancer initiation, cancer microenvironment, microRNAs, tumour progression, angiogenesis, cancer models

## Abstract

Cancer initiation and early tumour growth involve not only genetic mutations, but also important changes in the tumour microenvironment. The extracellular matrix (ECM) in particular undergoes biochemical and mechanical remodelling, influencing cell polarity and abnormal vascular development to drive cancer initiation and tumorigenesis. MicroRNAs (miRNAs) further regulate these early processes by altering gene expression, promoting invasion, and contributing to tumour plasticity. Recent advances in *in vitro* modelling have aimed to replicate tumorigenic environments using interdisciplinary approaches to better recapitulate *in vivo* environments. This review discusses how ECM mechanics, cellular polarity loss, and miRNA dysregulation collectively contribute to tumour progression. It also highlights how current/emerging *in vitro* models, including self-assembling peptide hydrogels and bioprinting technologies, provide insight into strategies to recapitulate cancer initiation and early tumour growth.

## 1. Introduction

Cancer remains a leading cause of morbidity and mortality worldwide [1]. Its initiation and progression are driven by complex interactions between cancer cells and their surrounding microenvironment [2,3,4]. The extracellular matrix (ECM) is a dynamic, three-dimensional (3D) network of macromolecules that plays a pivotal role in modulating cellular behaviour, providing structural support, and orchestrating biochemical/mechanical signalling. These cell–matrix interactions are crucial for tissue organisation and homeostasis [5,6]. Alterations in the ECM significantly contribute to various aspects of cancer, including initiation and early tumour growth, by promoting the formation of malignant environments. These environments can stimulate changes in cell polarity, dysregulate micro RNA (miRNA) expression, and promote vascularisation to aid tumour development [6,7,8].

A fundamental hallmark of early tumour development is the disruption of epithelial cell polarity. Its disruption can lead normal tissue to transform into a cancerogenic phenotype. The loss of apical/basal polarity, mediated by key polarity complexes such as Crumbs, Par, and Scribble can distort cellular adhesion and communication [9,10,11]. These molecular changes not only compromise tissue homeostasis but also prompt processes like the epithelial-to-mesenchymal transition (EMT), which are key in tumour initiation and progression. EMT process have been known to contribute to tumour plasticity, promoting cancer stem cell (CSC) formation and supporting tumour vascularisation, all important facets of tumour development/maintenance [10]. However, in recent years, miRNAs have also emerged as key regulators of ECM remodelling and cancer progression, influencing gene expression networks that govern cell adhesion, migration, and invasion [8,12]. Their dysregulation has been shown to significantly influence both tumorigenesis and tumour suppression. MiRNA signalling pathways in cancer have been shown to promote ECM degradation, angiogenesis, and immune evasion. Notably, miRNAs such as miR-200 family, among others, have been identified as central players in early tumour pathophysiology, making them promising targets for therapeutic intervention [13,14].

As tumours progress beyond initiation, they develop mechanisms to sustain their growth, particularly through the formation of new vascular networks. Neovascularisation, driven by vasculogenesis and angiogenesis, ensures adequate oxygen and nutrient supply to proliferating malignant cells. While angiogenesis arises from pre-existing blood vessels, vasculogenesis involves the recruitment of endothelial progenitor cells from bone marrow [15,16,17]. The interplay between these processes, mediated by hypoxia-induced signalling and ECM modifications are integral in during cancer initiation tumour development and resistance to conventional therapies [18,19].

Given the complexity of these interactions, *in vitro* models have been developed to recapitulate the tumour microenvironment and investigate early cancer pathophysiology. Advances in three-dimensional culture systems, microfluidic platforms, and biomimetic scaffolds have provided valuable insights into ECM–cancer cell dynamics [4,20,21]. Thus, offering new opportunities for better understanding the mechanisms involved during dynamic process such as cell polarity alterations, dysregulation of miRNAs, and malignant vascular development. Furthermore, they can be used as suitable tools for drug discovery, translational research and personalised medicine [22,23,24].

This review aims to explore key mechanisms responsible for ECM remodelling, cell polarity disruption, and vascular development during cancer initiation and early tumour growth. By integrating current knowledge from molecular biology, biomechanics, and bioengineering, we highlight the critical role of the ECM in shaping tumour progression and discuss emerging *in vitro* modelling strategies for studying these aspects of cancer pathophysiology as well as their potential for the development of therapeutics.

## 2. ECM Remodelling in Cancerogenic Environments

The extracellular matrix is formed of around 300 proteins including fibrillar and non-fibrillar collagens, proteoglycans and glycoproteins including laminins, fibronectin and elastins, although its exact composition is tissue-specific [25,26]. Glycosaminoglycans (GAGs) bind with proteoglycans, maintaining them across the collagen fibrils and retaining growth factors within the ECM [25]. The ECM not only provides anchorage for cells, but chemical and mechanical signals from the ECM mediate a number of cell functions that influence cellular behaviour and morphology [26,27].

During tumorigenesis, the highly dynamic ECM is remodelled through increased matrix metalloproteinase (MMP) activity, collage/hyaluronic acid deposition and fibre cross-linking with associated reduction in matrix pore size and linearisation of collagen fibres [28]. This promotes a stiffer microenvironment that alters the chemical and mechanical signals cells receive from the ECM. The stiffer matrices stimulate abnormal cell behaviour, facilitates intravasation, drives tumorigenesis, and later, promotes metastisis [27,28]. Several studies have observed a correlation with tissue stuffiness and cancer development. Itoh et al., [29] observed that pancreatic tissue in healthy volunteers was ~2 kPa, whereas those with pancreatic cancer had a stiffness of ~6 kPa. Similarly, a comparative study of breast tissue obtained from 362 women diagnosed with breast cancer compared with tissue obtained from 656 healthy women demonstrated a positive association between breast tissue density and breast cancer risk [30]. In mesenchymal high-grade serous ovarian cancers, Mieulet et al., [31] demonstrated that not only was tumour stiffness positively correlated with tumour growth, but that tumour stiffness was correlated with increased collagen density, thickening and lengthening of collagen fibres. It also corresponded with an increase in myofibroblast content and activation of the mitogen-activated protein kinase (MAPL) pathway, illustrating the relationship between ECM remodelling, tumour aggressiveness and poor clinical prognosis.

Other studies, with increasing evidence, have also demonstrated that matrix stiffness has a significant role in cancer initiation. Chaudhuri et al. [32] reported that increased matrix density led to a change of non-malignant mammary epithelial cells towards a malignant phenotype through activation b4 integrin activation of the phosphoinositide 3-kinase (PI3K) pathway. Wood et al. [33] utilised a mechanically tuneable, alginate/Matrigel model to demonstrate that in mammary epithelial cells, increased matrix stiffness alters metabolic processes inducing DNA damage in a RhoA-dependant manner. Furthermore, You et al. [34] observed that matrix stiffness had the potential to initiate stemness in cancer cells through the b1/Akt/mTOR/Sox2 signalling pathway. These studies not only showed that ECM stiffness has a crucial role in cancer initiation, but also significantly influences cell behaviour and function. Therefore, the ECM is significantly implicated in various mechanisms of cancer including cell polarity and neovascular development.

Apart from matrix stiffness, mechanical forces such as shear stresses or fluid dynamics can significantly influence changes to the ECM and play a role in early tumour growth [35,36]. As solid tumours grow, they have been shown to accumulate mechanical stresses, in part due to proliferative cancer cells, but also due to radial and circumferential stresses from surrounding tissues. The accumulation of solid stresses can lead to the compression of intratumoral blood vessels and generate hypoxic cores within tumours. Such environments are frequently implicated in the induction of angiogenesis through hypoxia-inducible factor 1-alpha (HIF-1α) signalling, which facilitates the delivery of oxygen and/or nutrients to hypoxic cells, while concurrently hindering the effective distribution of chemotherapeutic agents due to compromised vascular perfusion. Both phenomena interlink to drive tumour progression [35,37]. The upregulated signalling of HIF-1α, induced by shear stresses within hypoxic environments, has been associated with the metabolic reprogramming of cells in tumorigenic environments, primarily by enhancing their glycolytic activity. This, in turn, promotes the growth, survival, proliferation, and long-term maintenance of tumorigenic environments [38,39]. However, mathematical models, such as those proposed by Mpekris et al. [37], have demonstrated that spatial heterogeneity exists within tumours. Specifically, while the tumour core experiences an accumulation of compressive solid stresses, tensile forces are more prominent at the periphery. This heterogeneity in tissue mechanics was shown to correlate with increased vascular density and oxygenation at the tumour margins, which in turn corresponded with higher proliferation rates of cancer cells compared to the core of tumours [37,40]. Furthermore, studies such as those by Lee at al. [41] and Hyler et al. [42] have shown that low levels of fluid shear stress can activate YAP/TAZ signalling pathways to drive cancer cell motility. Additionally, such mechanical cues can induce structural and/or genomic alterations in benign cells, facilitating their transformation towards a cancerous phenotype. These findings highlight the importance of mechanotransduction signals in tumour development and progression.

## 3. Epithelial Cellular Polarity and Early Tumorigenesis

In healthy tissue, epithelial cellular polarity is fundamental for the maintenance of tissue homeostasis [43]. It establishes the directional movement of proteins between the apical and basal domains required for the correct localisation of ion channels, signalling receptors and ECM components, all important for intracellular signalling. Furthermore, it mediates tissue organisation through regulation of the division plane position during mitosis [7,44,45,46].

Epithelial cells can be defined by three distinct domains: the apical domain, which faces the lumen; the basal domain, which is in direct contact with the basement membrane through integrin receptors; and the lateral domain, which expresses tight junctions for adhesion to neighbouring cells [47]. In the apical domain, the Crumbs complex through Crumbs cell polarity complex component 3 (Crumbs3), Pals1 and Pals1-associated tight junction protein (Patj) are crucial to regulate epithelial cells polarity, tight junction integrity, and maintain epithelial homeostasis [48]. These proteins interact with Par complex proteins and enzymes such as partitioning defective 3 (Par3) and 6 (Par6), atypical protein kinase C (aPKC), and cell division cycle 42 (Cdc42) to preserve apical domain. In the basal domain, the Scribble complex, formed of Lethal 2 giant larvae (Lgl), Scribble and Discs Large (Dlg), acts as an apical polarity antagonist factors to help cells maintain cell polarity by excluding them from the basal domain [47]. Scribble acts independently to maintain Epithelial cadherin (E-cadherin) adhesion junctions, while Lgl has roles in the trafficking of proteins to the basal domain as well as acts as an antagonist to aPKC to maintain cell polarity [49]. The lateral domain forms a physical intercellular barrier that controls the diffusion of ions/small molecules across the epithelium and segregates the apical and basal domains [46,48,50]. Formed of Claudins, Junction adhesion molecules (JAM) and zonula occludins have a reciprocal relationship with polarity proteins where they act as scaffolds for polarity complexes which in turn mediate the assembly of tight junctions [51].

Due to the critical role cell polarity plays in tissue organisation, cell signalling, and cell function, the dysregulation of key interactions associated within the distinct domains in epithelial cells can be detrimental to cell/tissue homeostasis. These disruptions have been linked to various aspects of disease and have been associated with mechanisms of early tumorigenesis in cancer (Figure 1). Therefore, comprehending the underlying causes of these disruptions can aid in our understanding of cancer initiation as well as tumour progression and find new therapeutic targets.

### 3.1. Dysregulation in Cellular Polarity During Cancer Initiation and Early Tumorigenesis

Altered cell polarity is often considered a hallmark of carcinogenesis. Phenotypically, tumours have been shown to exhibit either a partial or complete loss of polarity or altered polarity such as inverse polarity and loss of expression of polarity proteins. However, this phenotype is dynamic, with cancer cells being able to gain polarity during metastasis [10]. Altered polarity has been directly linked with disruption to the tumour suppressor and oncogene signalling pathways. This can lead to increased cell proliferation, altering nucleus positioning, and multilayering or filling of the luminal space, resulting in a loss of tissue architecture, as well as promoting cancer cells towards an epithelial-mesenchymal transition (EMT) [10,56].

An example where dysregulation in cell polarity can lead to cancer initiation and accelerate progression was reported by Mescher et al. [9]. It was observed that the loss of Par3 in epidermal keratinocytes promoted proliferation and dedifferentiation as a result of the upregulation of P-cadherin. High P-cadherin is associated with reduced survival rates in malignant melanoma cases. Furthermore, in prostate cancer, a loss of Par3 has been associated with hippo signalling inhibition and a change in mitotic spindle orientation and multilayering [57]. Moreover, the loss of Par3 has also been linked to the development of high-grade intra-epithelial neoplasia, a pre-malignant multilayered structure from liver luminal and basal epithelial cells [10,44,56].

Other polarity proteins, such as Lgl and Scribble, have been shown to be involved in breast, prostate, lung and colorectal cancers when altered. Lgl has been shown to be upregulated in estrogen receptor-positive breast cancers, while Scribble is downregulated and mislocalised in mammary tumorigenesis [58,59]. In prostate and bladder cancers, the loss of Dlg isoform has been observed, and Feigin et al. [11] suggested that Scribble, when mislocalised, has a role in mammary tumorigenesis. Using transgenic mice models expressing a Scribble mutant that failed to localise cell–cell junctions, it was noticed that high degrees of Scribble mislocalisation could promote mammary tumorigenesis via interactions with Phosphatase and tensin homolog (PTEN) and activation of the Akt/mTOR/S6kinase signalling pathway. Thus, demonstrating that the dysregulation of key polarity junction proteins can significantly drive tumorigenicity and a deeper understanding of how the dysregulation of these proteins is initiated *in vivo* are needed.

### 3.2. In Vitro Models to Recapitulate Cell Polarity Dysregulations in Cancer

In recent years, *in vitro* 3D cell culture models utilising hydrogel models have become the standard for understanding cell polarity and its role in tumorigenesis. They are able to overcome limitations often attributed to conventional two-dimensional (2D) monolayer platforms that lack the organisation, morphology, cell–cell and cell–matrix interactions seen in the 3D and *in vivo* environment [60,61]. Early modelling of cellular polarity utilised Madin–Darby canine kidney (MDCK) kidney cells [62]. Work such as that of Yu et al. [63] deepened our understanding of cellular polarity, showing how cell orientation and regulation differs in 3D using collagen hydrogels. Thereafter, others have attempted to model other types of epithelial tissue and observed that cellular polarity is intrinsically linked to tumorigenesis. Work from researches such as Zhan et al. [60] used Matrigel™, a reconstituted basement membrane derived from Engelbreth–Holm–Swarm mouse sarcoma, to better understand the mislocalisation of polarity proteins. It was observed that proteins such as Scribble are associated with both a loss of apoptosis and tumour initiation. Other researchers such as Chatterjee et al. [64] took a combinative approach using the mechanical properties of collagen I with the promotive properties of Matrigel™ to form hydrogels representative of ‘soft’ and ‘stiff’ extracellular environments. MCF10A non-cancerous breast epithelial cells cultured on stiffer collagen I containing matrices exhibited a loss of polarity with phenotypic plasticity like those seen in oncogene-naïve epithelial cells and unlike those cultured in only Matrigel™ matrices. It further demonstrated that they displayed a more invasive behaviour than those cultured in softer Matrigel™ matrices. Thus, demonstrating matrix stiffness plays a crucial role in mediating cell polarity and may potentially influence non-cancerous cells to behave like or switch into a cancerous phenotype.

#### 3.2.1. Advances in Hydrogel in Vitro Modelling to Study Cell Polarity

The use of naturally derived matrix components such as Matrigel and collagen are limited by batch-to-batch variability and poor tunability. As a result, in recent years, the focus has shifted into developing novel biomaterials that overcome these limitations. Owing to its novelty, the focus has been on developing materials that support the formation of epithelial tissues that exhibit polarity, and can then be manipulated to deepen our understanding of polarity and tumorigenesis [65,66]. Work from researchers such as Biderra et al. [67] utilised naturally derived bioinert hydrogels. Alginate functionalised with Arg-Gly-Asp (RGD), a short attachment peptide motif found in fibronectin, laminin and vitronectin, was used to generate more chemically well-defined hydrogels. These hydrogels promoted the formation of biologically relevant acini displaying apical–basal polarity, lumen formation and basement membrane deposition of non-tumorigenic mouse mammary EpH4 epithelial cells. Furthermore, EpH4 cells were observed to transition into mesenchymal-like phenotypes with inside-out polarity, demonstrating the potential of their model to replicate aspects of tumorigenesis.

Other approaches in recent years have moved towards fully synthetic ECM mimics. Nowak et al. [68] created a fully synthetic hydrogel using a Heparin and multiarm star Poly(ethylene glycol) (PEG) or starPEG. Hydrogels were functionalised with an MMP-cleavable sequence to promote the formation of mammary acini. MCF10A cells cultured in these functionalised synthetic matrices were polarised and observed to have greater matrix deposition and could form more lumen than non-heparin-containing PEG–PEG gels. A different study by Weiss et al. [69] compared the effectiveness of cell adhesive motifs RGD and Tyr–Ile–Gly–Ser–Arg (YGSIR) in a PEG-based hydrogel to influence cell morphology and behaviour. MCF10A, a non-malignant epithelial cell-line, maintained a phenotypical epithelial cell morphology with less invasive behaviour in the presence of cell adhesive motifs in gels. In contrast, MCF10A cells overexpressing ErbB2 to promote a pre-malignant phenotype and malignant triple-negative breast cancer epithelial cells MDA-MB-23 both had altered morphology and displayed more proliferative/invasive behaviour. More recently, Zhang et al. [70] developed oligo(ethylene glycol)-grafted polyisocyanates or PIC hydrogel functionalised with RGD. These hydrogels were able to support proliferation and polarised organisation of dissected mouse mammary gland fragments in culture to form mammary gland organoids, thus showing potential to be implemented for generating tumorigenic organoids from cancerous tissue or cells.

More novel approaches to model cancer pathophysiology and polarisation have introduced self-assembling peptide hydrogels (SAPHs) to mimic ECMs. SAPHs formed of alternating hydrophilic and hydrophobic amino acid residues can self-assemble into a fibrinous network mimicking collagen. Clough et al. [71] demonstrated that SAPHs could be used to model early-stage breast cancer. Using MCF7 and MDA-MB-23 cells, cells were encapsulated within a neutrally charged SAPH to generate microenvironments that mimicked key feature of solid tumours including hypoxia and invasive behaviour. This was dependent on the cell type as MCF7s formed large spheroids resembling acini, whereas MDA-MB-231 displayed their phenotypical dispersed characteristics, therefore demonstrating that SAPH-based systems could recreate *in vivo*-like conditions. Lingard et al. [72], on the other hand, used a negatively charged SAPH and showed it could promote the formation of polarised mammary acini through functionalisation with laminin. Furthermore, the study demonstrated that the mechanical properties of the hydrogel could be easily tuned to alter cellular gene expression profiles and cell viability within the matrix, showing potential use for recapitulating different aspects of tumour development just by simple modifications to matrix characteristics.

#### 3.2.2. Alternative *In Vitro* Modelling Strategies of Cellular Polarity and Tumorigenesis

Alternative methods of *in vitro* modelling, such as bioprinting, have been more prominently used in recent years. Bioprinting allows the precise localised deposition of cells and growth factors for the formation of tissue or tissue-like constructs with complex architecture [73]. Swaminathan et al. [74] demonstrated the usefulness of bioprinting in understanding breast cancer at different disease stages. MCF10A cells and cancerous or cancer-induced (MCF10A-NeuN, MDA-MB-231 and MCF7) cell spheroids were bioprinted in Matrigel, gelatin-alginate and collagen-alginate bioinks. Bioprinted MCF10A and MCF10A-NeuN spheroids maintained their pre-printing morphology after 72 h, whereas spheroids printed form cancerous cell-lines (MCF7and MDA-MB-231) developed an invasive behaviour representative of their metastatic breast cancer, suggesting a potential change in cell polarity. Creff et al. [75] developed a PEG-DA and acrylic acid hydrogel that combined stereolithography bioprinting to produce constructs with small intestine crypt/villus-like topography. When seeded with Caco-2 intestinal epithelial colon adenocarcinoma cells, hydrogels mediated the formation of a polarised epithelium mimicking aspects of the *in vivo* environment seen in intestinal tissue. In another alternative biofrabrication strategy, Fischer et al. [76] developed a nanotopographic ECM-coated substrates that mimic the collage fibre size and waveforms seen in tumours *in vivo*. The effect of ECM fibril architecture demonstrated that the smaller peaks on linear substrates, which mimicked tumour ECM remodelling, could direct cancer cells to orient through stress fibres and drive migration along the wave axes via focal adhesions cues. However, cells grown on wavy substrates, stress fibres, and focal adhesion on cells was shown to be depolarised, inhibiting migration. These approaches demonstrate how new methods of replicating the *in vitro* tissue architecture allows for a better understanding of cellular polarity, early tumorigenesis and tumour development.

## 4. Cancer miRNAs and Their Impact on Tumour Initiation, Progression and ECM Remodelling

MiRNAs are small, around 18–25 base pairs, negatively charged, non-coding RNAs, with a cross-section of ~2.5 nm and length ~6–7 nm. Nucleic base pairing interactions between pyrimidines and purines have a fundamental role in how miRNAs regulate gene expression by binding to target messenger RNAs (mRNAs) to repress translation or degrade binding mRNA [77]. Initially, little interest was given to miRNAs when discovered in the 1990s. However, work by Ambros and colleagues [78,79] showed how they affect the genes lin-4 and lin-14 in posttranscriptional regulation during protein synthesis [80]. This demonstrated that several physiological and developmental processes are influenced by miRNAs due to their ability to control the gene expression of target mRNAs. The biogenesis and processing of microRNAs are depicted in Figure 2.

As post-transcriptional regulation of protein synthesis occurs in all cells, miRNA profiles are ever-present. MiRNAs have attracted a lot of research efforts as indicators for neurodegenerative, cardiovascular, autoimmune diseases, and cancer [81]. In many types of cancer, miRNAs modulate key processes, such as metastasis, apoptosis, proliferation, or vascular development [82]. Based on the mRNA regulated, they can act as oncogenes (genes that have the potential to transform a cell into a tumour cell) or as tumour suppressor genes (genes that maintain healthy cell division). Numerous miRNA cancer biomarkers have been studied over the years (Table 1) for both prognostic and diagnostic purposes [83]. However, many common miRNA biomarkers are non-specific, as they are associated with various forms of cancer like miR-141, which is a key prostate cancer biomarker but is observed in many other epithelial cancers, breast, colon, and lung [84]. Nonetheless, despite the lack of specificity, they have been shown to be great prognostic tools. In fact, due to their lack of specificity, they are considered great therapeutic targets for multiple types of cancer with heterogenous characteristics. Yet, equally, they can be used to model or study different types of cancer *in vitro*.

Biomarkers from the MiR-200 family (including miR-200a, miR-429, miR-200c and miR-141) have been widely studied in part because they are significantly conserved in vertebrates but also highly expressed in epithelial cells during cancer initiation/metastasis [13,85]. They have been shown to target vascular endothelial growth factor A (VEGFA) [86], as well as SIRT1, TSP-1, and c-kit. In contrast, miRNAs such as miR-16 and miR-29 can be anti-angiogenic by inhibiting VEGF expression and suppressing tumour growth [87,88,89].

MiR-155 is also overexpressed during carcinogenesis [14], particularly in breast cancer. It has been shown to have prognostic value in breast cancer by targeting genes [83,84,85,86,87,88,89,90,91,92,93,94] such as RhoA, which regulates cellular polarity and tight junctions; thus, the suppression of this factor promotes EMT [95].

Similarly, miR-21 is well known to be upregulated in almost all types of cancers, mediating various mechanisms of cancer [96] and contributing to tumour progression by suppressing tumour suppressor genes such as PDCD4 and PTEN, while also enhancing cell proliferation and survival [82,97,98,99,100,101].

The dysregulation of miRNA expression also influences ECM remodelling, a key feature of tumour microenvironment. MiRNAs can target ECM-regulating proteins, such as MMPs and their inhibitors to change the matrix [8]. The production of collagen, which is significantly abundant in the ECM surrounding tumour cells, can also be significantly altered by miRNAs. MiRNAs let-7g, miR-29b and miR-29c target multiple collagens and influence the ECM stiffness [12]. Other miRNAs, like miR-133a, can bind to collagen type I alpha 1 (Col1A1) to promote collagen accumulation, which in turn generates more attachment sites to facilitate cancer invasion [102].

Certain miRNAs, such as the miR-200 family, miR-21, and miR-155, can function as tumour suppressors or oncogenes, adjusting the molecular environment of early tumour formation. Others, such as miR-181b or miR-29b/c, have a greater influence in remodelling the ECM of tumorigenic tissue. To better understand the various mechanisms of cancer initiation and tumour development, miRNAs have been used in cancer modelling to mimic various aspects of cancer pathophysiology or better study miRNA profiles found in a wide range of cancers.

### 4.1. MicroRNA in Cancer and Cancer Models

Significant research has focused on miRNA-based therapeutics to examine pro- or anti-tumoral effects. Various modelling strategies have been developed over the years to identify miRNAs as oncomiRNAs or tumour suppressors [103,104]. Tumour-growth-enhancing studies such as those by Dal Xiao et al. [105] reported that the overexpression of miR-17-92, a polycistronic gene often amplified in lymphomas, could enhance the proliferation/survival of lymphocytes via the suppression of PTEN and Bim. By suppressing these proteins, miR-17-92-transgenic mice developed lymphoproliferative disease and autoimmunity. Conversely, loss of function studies such as those by Wu et al. [106] showed that let-7, a key regulator of stemness and regeneration, could act as a tumour suppressor. By overexpression of this miRNA, the suppression of tumour activity was observed in MYC-driven hepatoblastoma models. Additionally, it was observed that excessive let-7 expression compromised tissue protection and impaired liver regeneration, suggesting that a balanced let-7 level is critical for cancer suppression without harming normal tissues. However, other studies, like those by Li et al. [107] and Banyard et al. [108], overexpressed the miR-200 family to study the mechanisms of EMT and their involvement in metastatic breast/prostate cancer mouse models. Therefore, as these studies show, a wide range of miRNAs are involved in regulating key features of cancer and tumour development. As a result, various *in vitro* modelling strategies have been employed to identify and profile key miRNAs involved in cancer pathophysiology to either gain a greater understanding of oncogenic mechanisms or determine potential therapeutic targets.

#### Three-Dimensional *In Vitro* Models to Study MicroRNAs Associated with Cancer Initiation and Growth

A common method used to study miRNA expression in tumour-like 3D environments was developed by Debnath et al. [109]. To generate 3D tumour organoids, Debnath et al. [109] coated surfaces with Matrigel^TM^ and waited until they solidified. Thereafter, cell suspensions in growth media supplemented with Matrigel^TM^ were seeded on top to generate 3D spheres. This study aimed to develop a protocol to recapitulate aspects of *in vivo* morphogenesis and oncogenesis *in vitro*. Since then, others have used this method or adapted it using other basement membrane extracts/components to generate 3D cancer organoids for studying cancer associated miRNAs [110].

Studies using basement membrane extracts like those by Nguyen et al. [111] demonstrated breast cancer epithelial cells (MDA-MB-231 and MCF7) had significantly different miRNA profiles when cultured in 3D using Matrigel^TM^ systems compared to conventional 2D platforms. The differential expression of 49 and 28 breast cancer morphogenesis-associated miRNAs were reported in MCF7 and MDA-MB-231 cells, respectively, when comparing 3D to 2D culture systems. This included the overexpression of EMT-inhibiting miRNAs (miR-141 and miR-429) in MCF7 cell lines in 3D culture compared to 2D. Interestingly, it was also observed that miR-429 was over expressed in MDA-MB-231 cell lines under 3D culture, which correlated with a less phenotypical elongated morphology commonly associated with that cell line. Salinas-Vera et al. [112] also compared 3D to 2D culture systems using Geltrex basement membrane extracts to generate *in vitro* 3D platforms and also observed significant differences between the two. Low expression of tumour suppressive miRNAs (miR-935, miR-195 and miR-140) was observed in triple-negative Hs578T breast cancer cell lines cultured in 3D compared to 2D. Similar low expression was also observed in breast tumour subtypes, thus providing more evidence that 3D platforms better mimic tumorigenic *in vivo* environments to a closer degree. Lõhmussaar et al. [113] used a Cultrex^®^ basement membrane extract embedded with cervical cancer patient-derived squamous carcinoma (SqCa) and adenocarcinoma (AdCa) cells to generate cervical tumoroids. The cell-laden solutions were seeded as droplets that solidified to generate 3D cell-laden spheres that developed into tumoroids once in culture. Tumoroids were observed to have a higher expression of Human Papillomavirus (HPV) viral encoding miRNAs compared to healthy organoids derived from non-cancerous cervical epithelial cells, thus providing important insight into the role these viral miRNAs have on tumour development.

While basement membrane extract strategies have provided great insight into the important role ECM components and 3D platforms have in recapitulating more *in vivo* like settings, similar environments have been produced using alternative components to study miRNA profiles/functions. Thippabhotla et al. [114] used a similar strategy to generate 3D culture environments to those using basement membrane extracts but used a synthetic, self-assembling peptide system (PepGel PGmatrix) instead. Although composed of synthetic components, significant differences were observed in cervical cancer Hela cells cultured in these 3D platforms compared to 2D. It was observed that the profile of extracellular vesicle (EV)-derived miRNAs, which play a crucial role in cancer initiation/progression, secreted by these cells had ~96% similarity with cancer patient plasma-derived EV miRNAs, whereas 2D cultures only correlated with ~80%. Similarly, a recent study by Godbole et al. [115] also reported significant differential miRNA expression of EV-derived miRNAs in their 3D culture platforms compared to 2D culture. However, they used photo-polymerised gelatin methacryloyl (GelMA) hydrogels laden with ovarian cancer cells lines (SKOV-3 and OVCAR-3) instead. Most notably, hsa-miR-30d-5p was the highest expressed miRNA in their 3D cultures, and elevated expression of this miRNA has also been strongly linked to early-stage ovarian cancer. However, simple salt-leaching casting methods have also been used to examine miRNA profiles in 3D environments like those developed by Balachander et al. [116]. Compared to 2D cultures, MDA-MB-231 grown on their 3D PCL scaffolds displayed significantly high upregulation of miR-210-5p and miR-146a-5p, two miRNAs associated with tumour initiation and the development of triple negative breast cancers, respectively, thus further demonstrating that cell behaviour can significantly change in 3D even in more simplistic approaches.

Collectively, these studies support that 3D environments better recapitulate the *in vivo* experience, and that the matrix significantly regulates miRNA expression. With a greater understanding of this, future models have the potential to introduce miRNAs into their systems and generate more well-defined microenvironments *in vitro*. This can significantly improve modelling strategies to develop cancer models with desired characteristics driven specifically by miRNA–matrix interactions observed *in vivo* and in 3D *in vitro* platforms.

## 5. EMT and EET in Tumour Initiation and Growth

The EMT is a dynamic process that allows polarised epithelial cells to undergo various biochemical changes and adopt a mesenchymal phenotype [117]. This causes cells to migrate away from the epithelial layer in which they originated and aids in a variety of morphogenetic events during early embryonic development, and also plays a role in wound healing later in life [117,118]. However, in recent decades, this process has been strongly linked to the reprograming of cancer cells towards a more aggressive phenotype with increased stemness. The generation of these cancer stem cells (CSCs) via the EMT have been associated with tumour initiation, metastasis and therapeutic resistance, therefore playing a key role tumorigenesis and progression [119,120]. A subtype of the EMT, the epithelial–endothelial transition (EET), in recent years has been shown to also play a pivotal role in early tumorigenic growth and progression [121,122]. In this process, cancerous epithelial cells differentiate in cells with an endothelial phenotype and derive into channels like blood vessels that can be used as alternative sources of nutrient and oxygen supply to support tumour growth [123,124]. As a result, understanding these pathophysiological processes is of great interest for the development of new therapeutic approaches that can target EMT/EET mechanisms.

### 5.1. The Role of EMT in Cancer Initiation and Early Tumour Growth

During early embryogenesis, the EMT is involved in various process including the formation of the placenta, the primitive streak during gastrulation, and the neural crest [125,126]. The composition of the ECM during these processes is highly abundant in Fibronectin and Laminins (mainly Laminin-111, Laminin-511), which play a pivotal role in providing cell adhesion sites as well as facilitating cell migration [127,128,129]. However, structural proteins such as collagens (collagen I, III, IV and VI) and glycosaminoglycan (GAGs) like hyaluronan are also present. Collagens contribute to the stiffening of the ECM, which enhances Rho-ROCK and YAP/TAZ signalling to promote cell contractility and migration via mechanotransduction signals during tissue development. GAGs like hyaluronan, on the other hand, act as ligands to facilitate cell motility [130,131,132,133]. These modifications to the ECM as tissues develop have been strongly linked to the increased expression of MMPs by epithelial cells, which in turn degrade the local ECM to help release latent transforming growth factor-beta (TGF-β), a major inducer of EMT, and allow the migration of the newly formed mesenchymal phenotypic cells [133,134].

Similar changes to the ECM to those observed in embryogenesis and tissue development have been observed in tumorigenic environments to promote cancer initiation and early tumour progression [4,135,136]. Changes in stiffness as well as oxygen in tumorigenic environments can induce epithelial cells towards an EMT. Stiffer matrices and oxygen deprivation in tumour environments have both been observed to modulate epithelial cells towards a mesenchymal phenotype and significantly regulate TGF-β signalling, a crucial inducer of EMT related cancer initiation (Figure 3A) [137,138]. Structural ECM proteins associated with stiffening of the matrix in tumours such as collagen I/ IV have been shown to enable cancerous and non-cancerous epithelial cells to transition towards an EMT [135,139,140,141]. EMT-induced cells can further give rise to CSCs via the upregulation of transcription factors like Snail, Twist, and ZEB1. The increased stemness of these tumour-initiating cells has been linked with various interactions with local ECMs to remodel and spark the formation of new tumorigenic microenvironments [34,142]. Furthermore, the EMT has often been associated with enhancing the survival of cancer cells by the increased expression of aldehyde dehydrogenases (ALDHs) and promoting immunosuppressive features to evade attacks from natural killers (NK), both linked to tumour initiation/propagation [143,144,145].

Their ability to survive hostile environments through the development of these features enables them to endure hypoxic environments [122,141]. To adapt to these environments, EMT-induced cells take part in the formation of vasculature during early tumour growth or undergo an EMT process subtype, the EET (Figure 3B), to form pseudo vessels via the process of vasculogenic mimicry (VM) [122,147]. However, the EET process is a less extensively studied transition process and remains poorly understood, but nevertheless has an important impact in sustaining tumour growth [121,122,148].

### 5.2. Development of EET to Support Tumorigenesis Under Hypoxic Environments

While the role of EET in embryogenesis does not seem to be apparent, it may be due to a lack of research or underreporting. However, its association with tumorigenesis has been more established in recent years, as has its ability to form VMs that patterns similarities to the generation of embryonic vasculogenic networks [149]. As tumours develop, varying degrees of localised hypoxia is common [150]. Cancer-associated fibroblasts (CAFs) and CSCs have been shown to modulate the ECM by the secretion of collagen, fibronectin, laminin, and MMPs in hypoxic environments, which in turn stiffen the matrix [151,152]. These changes in the matrix induce cells towards an EMT/EET phenotype, as does the expression of Twst1/Snail via Par1 and HIF-1α activation, although various aspects of the upstream signalling of Twist1 remain unclear [123,148]. However, in recent years, the expression of Slug has also been reported to participate in the induction of EET and formation of VMs [149,153]. The interplay of cell–ECM interactions in these hypoxic conditions can promote EET-induced cells to upregulate Vascular endothelial cadherin (VE-Cadherin), increase Notch signalling, and enhance VEGF expression, all key factors involved in the organisation of vasculature. Due to the similarity that EET-induced cells have with endothelial cells (ECs), this enables them to generate vascular-like structures, VMs, to supply oxygen and nutrients to oxygen/nutrient-deprived malignant cells growing in tumorigenic tissue [122,148].

The formation of VMs via the EET has not only been shown to aid in tumour development, but their presence has been linked with aggressive types of cancer with poor prognosis [122]. It is also believed that these pseudo-channels provide similar support to cancer progression to actual vasculature and can also connect with host blood vessels to form part of their network [154,155], thus playing a critical role in tumour growth. Therefore, the EET may have an even more prominent role than previously thought, and contributions to tumorigenesis should be studied further, as these are often overlooked.

### 5.3. In Vitro Modelling Methods to Mimic Tumorigenic EMT and EET Processes

Early studies using conventional 2D *in vitro* models like those by Batlle et al. [156] have been pivotal in our understanding of the EMT and its role in tumorigenesis. Their study provided great insight into the role Snail has in the repression of E-cadherin transcription to mediate epithelial cells towards the EMT. It further cemented this link between the EMT and the initial steps of cancer development/tumorigenesis. Another study by Cano et al. [136] used conventional 2D models to further establish the importance of Snail in EMT mediation. These findings were further validated *in vivo*, demonstrating how many EMT mechanisms involved in embryogenesis, especially Snail signalling, also occur during tumorigenesis using mouse embryo and xenografted tumours models. However, various modelling strategies have been developed over the years to bridge the gap between conventional 2D *in vitro* and *in vivo* models and produce 3D models to recapitulate the *in vivo* setting.

#### 5.3.1. Three-Dimensional Modelling Platforms to Study EMT Mediation in Cancer

One of the most common methods to study EMT and stemness in 3D has been using *in vitro* 3D floating spheroids generated on non-adherent surfaces [157,158]. While these studies have provided great insight into EMT and stemness, they do not recapitulate cell–ECM interactions displayed *in vivo*, such as the physical/mechanical cues that native ECMs provide. Scaffold-based approaches offer more control of the environment to improve reproducibility and can better mimic physical aspects of tumorigenic microenvironments that mediate the pathophysiological mechanisms of cancer such as the EMT [157]. Methods such as those by Wang et al. [159] fabricate 3D breast cancer models using hydrogels comprised of methacrylated hyaluronic acid and gelatin composites to study the effects of hypoxia for promoting the EMT. While the role hypoxia has on promoting the EMT is well known, this study demonstrated it could be studied in an environment that more closely mimics the *in vivo* setting. A more intricate method to study EMT interactions by Pal et al. [160] used a hybrid approach using hydrogels and electrospinning. Via this strategy, a more fibrous network was produced, similar to fibrous networks displayed in native ECMs. Cell-laden GelMA hydrogels were infused and solidified within a Poly Lactic-co-Glycolic Acid (PLGA) electrospun fibrous network. MDA-MB-231 cells embedded in these scaffolds were shown to significantly upregulate EMT markers (Snail1, ZEB1, and Twist2) as well as downregulate E-Cadherin when compared to cells grown in PLGA electrospun scaffolds or GelMA hydrogels alone. It further showed that not only were EMT markers enhanced, but proliferation was also improved in hybrid scaffolds, thus demonstrating the importance that topographical/mechanical cues play in regulating cell functions. A more recent study by Szostakowska-Rodzoś et al. [161] used 3D bioprinting to compare differences in stem cell and EMT markers in 2D vs. 3D culture conditions. Alginate-based bioinks laden with MCF7 cells were printed in three stacking layers. Compared to traditional 2D models, E-cadherin in MCF7 decreased significantly in 3D-printed models, thus suggesting it is driven towards an EMT phenotype. It was also observed that E-cadherin decreased significantly compared to more conventical 3D spheroid floating models, and that E-cadherin plays an important role in keeping spheroids compact, therefore highlighting a potential limitation of 3D floating spheroid systems. Together, these studies demonstrate the importance of 3D modelling frameworks to better replicate *in vivo* interactions, as they can significantly differ from those in conventional 2D and 3D low adherent surface culture systems.

#### 5.3.2. Three-Dimensional Modelling Platforms to Study EET and VM Processes

As an understudied process, the different methods of studying EET are limited. While many aspects of the process have been uncovered over the last decades, *in vitro* models have mainly relied in 2D platforms or using Matrigel^TM^ coatings/scaffolds to investigate EET and their unique feature of generating VMs [162]. Studies such as those by Xiao et al. [123] used a Matrigel^TM^ platform to delineate differences between EMT and EET and demonstrated that PAR1 has a significant role in directing epithelial cells towards a more endothelial phenotype involved in the formation of VMs. Using a Matrigel^TM^, others, like Sun et al. [149], have studied the relationship between high slug expression and the upregulation of VE-cadherin in EET induction, while Cannell et al. [154] reported a similar relationship with transcription factor FOXC2 as a key driver of VMs under hypoxic environments. Some studies like those by Shuai et al. [162] and Maniotis et al. [163] have attempted to engineer platforms with more defined ECM components like collagen and fibrin to study these EET/VM interactions in 3D. However, compared to other mechanisms/processes related to cancer, more complex 3D platforms to mimic the *in vivo* environment have not been used. Therefore, there is a large gap between our understanding of the EET and its cell–cell/cell–matrix interactions during cancer pathophysiology.

## 6. Vascularisation and Early Tumour Growth

The development of functional vasculature is essential for the overall maintenance of cell health and crucial in organ/tissue development. As the earliest events of organogenesis, it is critical for the survival and function of living organisms by removing waste products as well as delivering oxygen/nutrients to cells [164,165]. Vasculogenesis and angiogenesis are the two main processes involved in the formation of nascent vasculature. The vascular networks formed through these processes are not only important in the development of tissue but also in maintaining the homeostasis of healthy tissue [166,167]. However, the initiation of neovasculature once tissue has reached full development can be indicative of early tumour growth and progression. Therefore, these two processes play pivotal roles in stimulating the early growth of dormant solid tumours by supplying malignant cells with oxygen and nutrients to aid their proliferation and survival [168,169].

### 6.1. Vasculogenesis in Cancer and Tumour Development

Vasculogenesis is the process of the de novo formation of blood vessels by the proliferation and differentiation of endothelial cell precursors, known as angioblasts and hemangioblast, to assemble nascent blood vessels [170]. While many aspects of the ECM during vasculogensis remain unknown, laminin, fibronectin, collagen, and heparan sulfate proteoglycans have been identified to interact with specific receptors of endothelial cell precursors to promote their differentiation into ECs and the organisation of de novo vasculature via VEGF pathways [170,171]. After developmental stages, vasculogenesis occurs less frequently and is more often associated with pathological conditions such as ischemia or tumour growth [168]. Vasculogenesis in cancer is believed to be mediated by endothelial progenitor cells (EPCs) or bone marrow-derived hematopoietic cells (Figure 4). In the tumour microenvironment, VEGF secreted by hypoxic tumour cells is a key driver of vasculogenesis initiation by mobilising high-expressing vascular endothelial growth factor receptor 2 (VEGFR2) EPCs from bone marrow [172]. However, other factors are secreted by tumorigenic microenvironments like chemokines C–C motif ligand 2 and 5 as well as the hypoxia-responsive chemokine stromal cell-derived factor 1 (SDF-1) [173,174].

While growing evidence suggests that vasculogenesis is involved in tumour growth, it remains unclear whether it is involved in early or late stages of tumour growth. A study by Folkins et al. [15] showed that SDF-1 signalling by ganglia CSCs was involved in the recruitment of bone marrow EPCs in the early stages of tumour formation. Research by Paprocka et al. [16] observed vasculogenesis in relation to the increased number of circulating EPCs of patients with early-stage endometrial cancer. However, others, like Spring et al. [174], reported that EPCs did not integrate in early-stage tumour lesions but were more prominent in the vasculature of late-stage tumours of RIP1-Tag5 mice. Nonetheless, these studies highlight the importance of vasculogenesis in tumour development, and further research is needed to have a clearer understanding of its role in tumorigenesis.

### 6.2. Angiogenesis in Cancer and Tumour Development

Angiogenesis consists of the formation of new vascular networks from pre-existing blood vessels (Figure 5). The process results from a cascade of events influenced by cell–cell and cell–ECM interactions that direct sprouting ECs towards generating new vasculature [171,175]. The activation of VEGF induces the vasodilation of existing vessels and mediates the ECs to start secreting gelatinase MMPs to degrade the basement membrane. During this process, MMPs release bioactive basic fibroblast growth factor (FGF) and VEGF sequestered within the ECM to simultaneously degrade the basement membrane and induce the proliferation/migration of ECs that invade the tunica media of blood vessels [176,177]. These motile ECs form new capillary networks and express platelet-derived growth factor B (PDGF-B) to recruit mural cells like pericytes and vascular smooth muscle cells (VSMCs) to stabilise the newly formed vessels [178,179].

In cancer, angiogenesis is often initiated by the overexpression of HIF-1α from hypoxic cancer cells in dormant or solid tumours. This “angiogenic switch” in early tumour development, as reported by Folkman et al., can start in pre-malignant hyperplasic lesions [17,180]. As tumours grow beyond 1–2 mm in diameter, the upregulation of HIF-1α increases to provide oxygen/nutrients to developing hypoxic areas. HIF-1α upregulation causes a cascading effect, where tumour cells and surrounding stromal cells start secreting pro-angiogenic molecules that stimulate the overexpression of pro-angiogenic factors (VEGF, FGF, PDGF-B, and angiopoietins) [18,19,181]. This causes the ECM to remodel and stimulate ECs to organise new vascular networks that aid tumour growth and, in later stages, metastasis [182,183]. In recent years, various strides have been made to recreate this environment *in vitro* to study these underlying mechanisms and better understand potential therapeutic targets to inhibit vascularisation in tumours [24].

### 6.3. Vascularised In Vitro Models to Recapitulate Cancer Agiogenesis and Vasculogenesis

Biochemical and mechanical signals significantly regulate tumour microenvironments, including the formation of neovasculature [184,185]. One of most common methods to study this has been using Matrigel^TM^ assays. Various studies using Matrigel^TM^ have shown that tumour cells and ECs can organise into tubules and form vascular-like networks [186,187]. However, these assays have been limited in their recapitulation of the tumour microenvironment, as they lack the 3D cellular organisation and specific cellular orientations seen *in vivo* [24,188]. In recent decades, significant efforts have been made to mimic these microenvironments *in vitro* to better understand cancer pathophysiology and develop tools that target neovascularisation in tumours. These include hydrogel, microfluidic, and 3D bioprinting strategies that have been able to produce more complex tumour environments than those observed in more conventional Matrigel^TM^ assays (Table 2) [189,190,191].

**Table 2 cancers-17-01675-t002:** Advantages and limitations of different modelling methods for vascularised tumour environments.

Method	Description	Advantages	Limitations	References
Matrigel™ Assays	Basement membrane extracts from Engelbreth-Holm-Swarm used to study vascular-like networks	- Easy to use and widely available - Allows rapid assessment of vascular-like network formation	- Lacks *in vivo* 3D cellular organisation - Its composition is undefined/heterogeneous, which can hinder reproducibility	[186,187,192]
Hydrogel Models	Uses natural or synthetic polymers to form 3D hydrophilic ECM-like matrices	- Better mimics 3D architecture of ECM than Matrigel™ - Can incorporate well-defined chemical and mechanical factors to study vascularised cancer models	- Still lacks full complexity of *in vivo* settings - Difficulty in controlling the exact architecture of natural hydrogels - Some synthetic hydrogels may not fully support cellular functions	[189,193,194,195]
Microfluidic Models	Use of organ-on-chip platforms to recreate vascularised tumour microenvironments	- Enables dynamic control of fluids, nutrients, and signalling molecules via perfusion mechanics - Precise control of microenvironments and perfusion systems	- Often simpler vascular architectures than those observed *in vivo* - Achieving physiologically relevant conditions can be challenging	[24,196,197]
3D Bioprinting	Uses bioinks to spatially organised cells and fabricate complex vascularised constructs	- Precise spatial control of cells and scaffold/matrix components - Can precisely deposit multiple cell types and ECM components during fabrication	- Current models are still relatively simple compared to *in vivo* - Fabrication of bioprinted constructs can be technically challenging and remains relatively costly	[169,198,199]

#### 6.3.1. Hydrogels Models for Tumorigenic Vascular Development

Hydrogels have been used over the years for the biofabrication of vasculature and vasculasied networks, as they can mimic various aspects of natural ECM, providing more *in vivo*-like 3D environments than those of conventional 2D culture [200,201]. One of the earliest *in vitro* models to study sprouting vasculature using hydrogels, by Nakatsu et al. [189], used human umbilical vein endothelial cells (HUVECs) embedded in fibrin gels. Dextran beads laden with HUVECs were embedded in these gels and co-cultured with fibroblasts in media supplemented with angiopoietin-1 (Ang-1). This demonstrated that fibroblasts and Ang-1 both had synergistic roles in promoting sprouting lumen formation from HUVEC beads. This method with fibrin hydrogels was then later adapted by Hernandez-Fernaud [193] to investigate the role of chloride intracellular channel protein 3 (CLIC3), a protein secreted by CAFs and cancer cells. It was able to show sprouting angiogenesis in a more cancer-like environment and that CLIC3 was indeed a key driver for promoting ECs to form nascent vessels. However, hydrogels using synthetic materials such as PEG have also been developed in recent years. Roudsari et al. [194] fabricated PEG-RGD hydrogels embedded with Lung carcinoma cell lines and HUVECs, demonstrating that pro-angiogenic paracrine factors secreted by lung carcinoma cells could enhance vascular network formation by HUVECs. Furthermore, it showed that a synthetic-based hydrogel could serve as a suitable matrix to study vascular development in tumorigenic environments.

#### 6.3.2. Microfluidics to Model Vascularised Tumour Environments

In recent decades, microfluidic organ-on-chip models have been developed to model cancer and have shown great promise in recreating multicellular architecture and tumour–cell interactions in the presence of vascularised networks or single channels [24,196]. Sobrino et al. [190] developed micro-tumours in the presence of a microfluidic platform lined with endothelial colony-forming cell-derived ECs (ECFC-ECs) isolated from cord blood. Colorectal cancer (SW620, SW480), breast cancer (MDA-MB-231, and MCF-7), and melanoma (MNT-1) cell lines were seeded in fibrin gels that embedded the functionally perfused microfluidic platforms. These vascularised microfluidic tumour microenvironments were not only used to investigate the metabolic functions of cancer cells in the presence of vascular networks, but also assessed the viability of multiple chemotherapeutic drugs in their role of regressing vascular formation via the targeting of VEGF and PDGF receptors *in vitro*. In a different approach, Truong et al. [202] fabricated a microfluidic device with three different channels that could interact with each other in close proximity. One channel was seeded with glioma stem cell (GSC), while the other two were seeded with HUVECs and stromal cells, respectively. The HUVEC-laden microvascular networks stimulated GSC migration and proliferation and increased stemness markers (Nestin, SOX2, CD44) in a similar fashion to that observed *in vivo*.

Indeed, microfluidic strategies have demonstrated their viability to study different stages of cancer with significant potential to be employed for drug screening assays. However, organ-on-chip microfluidic strategies generally model simple architectures to mimic vasculature unlike those seen *in vivo* [196,197]. Therefore, they still fail to recapitulate some of the more complex compositions of tumour growth but have the potential to fabricate fluidic networks with more complex architectures to mimic native vasculature.

#### 6.3.3. 3D Bioprinting to Model Vascularised Tumour Environments

With advances in 3D bioprinting, it provides precise spatial deposition and organisation of cells to design complex tissue constructs *in vitro*. This allows for the integration of numerous combinations of cell types and supporting matrices within a highly controlled spatial environment with the potential to fabricate more accurate representations of the *in vivo* tumour microenvironment [169,199]. An early study by Li et al. [203] showed it was feasible to generate structurally complex vascular-like networks via 3D bioprinting. Using a double-nozzle extrusion technique, hepatocytes and adipose-derived stromal cells (ADSC) were spatially distributed to form vascular-like networks that mimicked aspects of liver tissue. More recently, Cheng et al. [204] used a one-step 3D bioprinting process that could simultaneously print GelMA bioinks containing breast cancer epithelial cells (MDA-MB-231), HUVEC, and osteoblasts to develop complex vascularised tumour models. The models were able to replicate aspects of tumorigenic bone tissue, and the precise alignment of these cell types was also able to mimic tissue angiogenesis as well as processes of bone metastasis. Cui et al. [205] used a laser direct writing technique to generate a similar model to that of Cheng et al. [204], demonstrating the versatility of 3D bioprinting techniques to generate complex vascularised tumour-like structures that can mimic the trans-endothelial migration and colonisation of cancer cells *in vitro*.

Despite significant advances in generating more complex structures, so far, 3D bioprinting techniques still generate relatively simple models compared to the *in vivo* environment. However, it offers great potential, as bioinks can be loaded not only with cells, but also bioactive molecules that can be precisely deposited to further mimic ECM components, better representing the cell–matrix interactions observed *in vivo* [198,199].

## 7. Conclusions and Future Perspectives

The role of the ECM in cancer initiation and early tumour growth is multifaceted. It involves structural, biochemical, and mechanical changes that drive tumour progression. Disruptions in cellular polarity and miRNA expression contribute to the cancerogenic remodelling of the ECM, as well as the neovascularisation of tumours. Collectively, these factors contribute to the transformation of normal tissue into tumour-supportive microenvironments. Understanding these mechanisms during early-stage cancer provides crucial insights for developing targeted therapeutic strategies aimed at mitigating tumour initiation and progression. The integration of an interdisciplinary approach that combines molecular biology and engineering has great potential to advance our understanding of tumour–ECM interactions by bioengineering *in vitro* models that can recreate *in vivo* cancer environments.

The development of more sophisticated *in vitro* models that accurately mimic tumour microenvironments can help elucidate various mechanisms of disease currently unknown. Furthermore, it can enhance pre-clinical studies and accelerate the discovery of novel anti-cancer therapies. Future research should also focus on identifying novel biomarkers and therapeutic targets within the ECM that could serve as early diagnostic tools or intervention points. Targeting ECM-related pathways, modulating miRNA expression, and disrupting tumour–stromal interactions may offer new insights for precision medicine in oncology. Ultimately, a deeper understanding of ECM-driven cancer mechanisms will pave the way for innovative therapeutic strategies and potentially improve clinical outcomes.

## Figures and Tables

**Figure 1 cancers-17-01675-f001:**
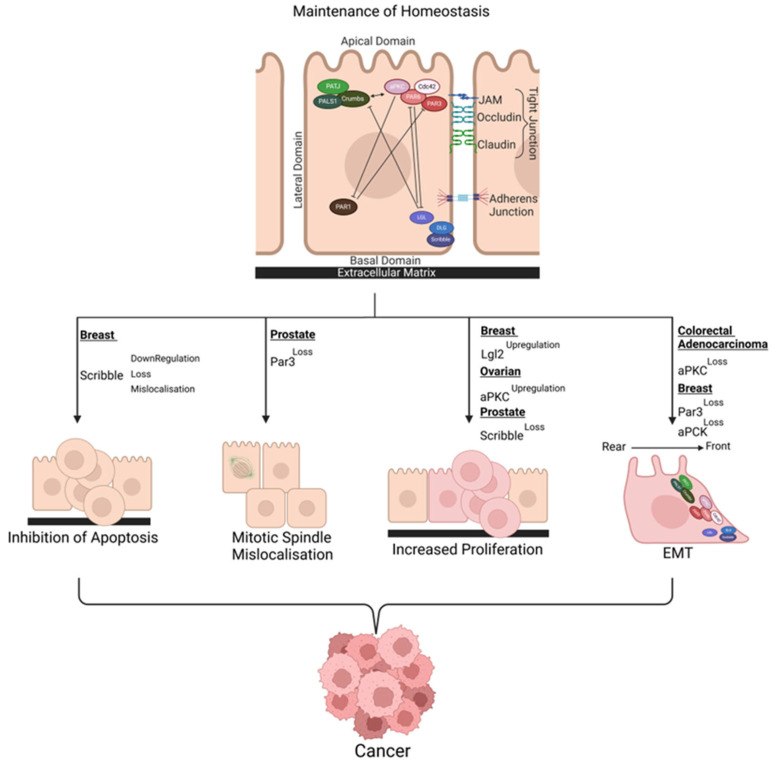
The polarity protein pathways during tissue homeostasis. The Par complex (red) localised to tight junctions, the Scribble complex (blue) at the basal edge and the Crumbs complex (green) in the apical domain. Par1 is not part of the three primary complexes, but is fundamental in excluding apical proteins from the basal edge [52]. Single-ended arrows indicate excusatory activity, whilst double-ended arrows indicate direct interactions. Examples of how specific polarity proteins are dysregulated in certain cancers and the downstream effects directly linked with polarity protein dysregulation are shown. Polarity proteins interact with a variety of cellular pathways in healthy tissue including Hippo, mTOR, Hedgehog, JAK/STAT and MAPK1 pathways and dysregulation of polarity proteins has downstream effects on these pathways leading to tumour-associated phenotypes [10]. The upregulation of aPKC in ovarian cancer, the loss of Scribble in prostate cancer, the loss of aPKC in Colorectal adenocarcinoma, and the loss of both Par3 and aPKC in breast cancer [53,54,55]. (Created with BioRender.com, accessed on 4 May 2025).

**Figure 2 cancers-17-01675-f002:**
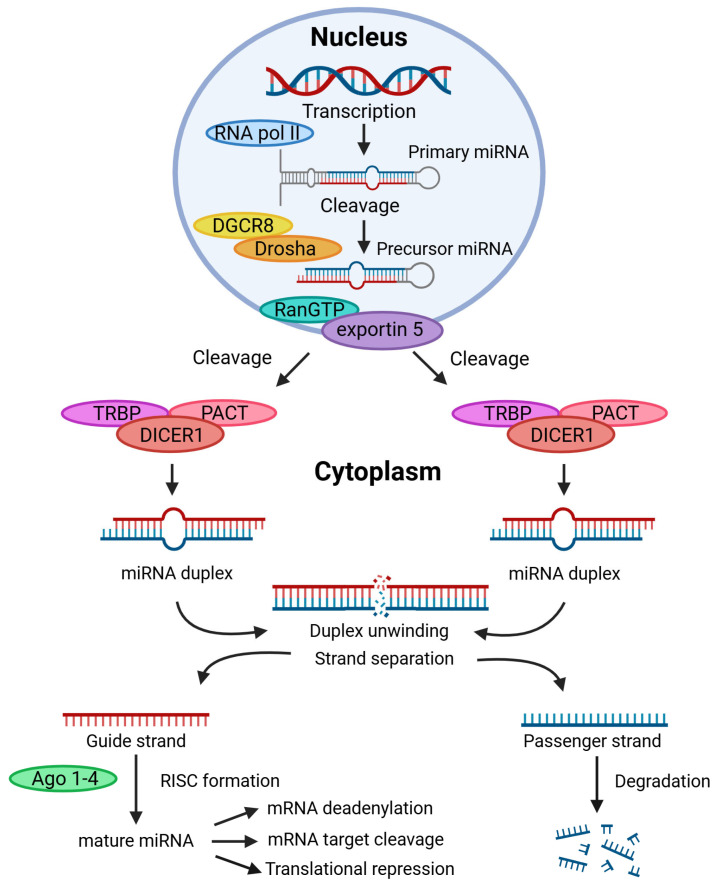
Schematic of microRNA biogenesis and processing. miRNA coding genes are transcribed by RNA polymerase II (RNA pol II) in the nucleus, producing primary miRNA. The primary miRNAs are further processed by the microprocessor complex containing the Dorsha and DiGeorge Syndrome critical region 8 (DGCR8) to generate 60–70-nucleotide precursor miRNA. The precursor miRNA is then exported from the nucleus to the cytoplasm by Exportin 5 in a RanGTP-dependent manner. In the cytoplasm, pre-miRNAs are further processed by the DICER1 complex (which includes cofactors such as TRBP and PACT) into an 18- to 24-nucleotide miRNA duplex. While the single-stranded passenger strand miRNA will be degraded rapidly, the other single-stranded miRNA of the duplex (the guide strand) is incorporated into the Argonaute (Ago1-4) protein with the RNA-induced silencing complex (RISC) to form mature single-stranded miRNA. Mature miRNA will be directed to its complementary target mRNA where they can cleave to, induce mRNA deadenylation, and mediate translational repression by binding to target mRNAs. (Created with BioRender.com, accessed on 4 May 2025).

**Figure 3 cancers-17-01675-f003:**
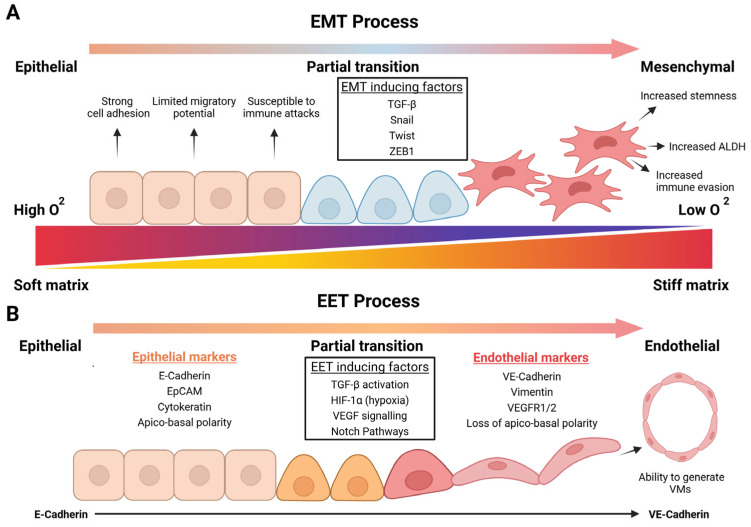
Schematic of epithelial–mesenchymal transition (EMT) and epithelial–endothelial transition (EET) processes, which contribute to cancer initiation progression and metastasis. The EMT process is characterised by the transition from epithelial cells towards a mesenchymal phenotype. The loss of polarity via EMT-inducing factors enhances cell motility, stemness potential, and immunosuppressive features [145]. The acquisition of these characteristics by EM-induced cells has been linked to the stiffening of the ECM and development of localised hypoxia in tumorigenic environments. The progressive gradient represents the gradual nature of EMT, transitioning from an epithelial state through intermediate hybrid phenotypes before fully acquiring mesenchymal characteristics (**A**). The EET process refers to the transformation of epithelial cells into endothelial-like cells. This transition is marked by the loss of polarity, the downregulation of epithelial markers and the acquisition of endothelial markers, such as VE-cadherin and CD31 by epithelial cells [122,146]. Through EET-inducing factors, the acquisition of endothelial features enables these cells to generate VMs that resemble vessel-like constructs with similar properties to those of blood vessels. This VMs have the ability to supply oxygen and nutrients to hypoxic cells in oxygen-deprived tumorigenic environments to sustain tumour growth (**B**). (Created with BioRender.com, accessed on 4 May 2025).

**Figure 4 cancers-17-01675-f004:**
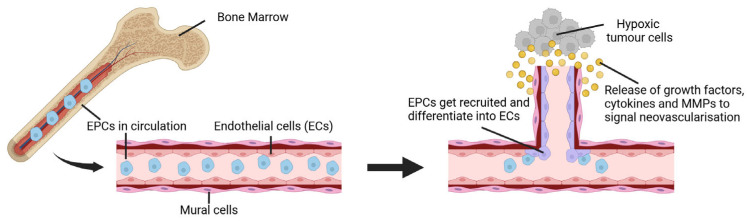
Tumorigenic neovascular development via the vasculogenesis process. Endothelial progenitor cells (EPCs) originate from the bone marrow and circulate the bloodstream. Hypoxic tumour cells secrete cytokines and pro-angiogenic growth factors to signal EPCs recruitment as well as induce their differentiation into endothelial cells (ECs). Differentiating ECs are also signalled to secrete MMPs to degrade surrounding ECM and mobilise towards hypoxic tumorigenic environments, eventually assembling into new blood vessels. (Created with BioRender.com, accessed on 4 May 2025).

**Figure 5 cancers-17-01675-f005:**
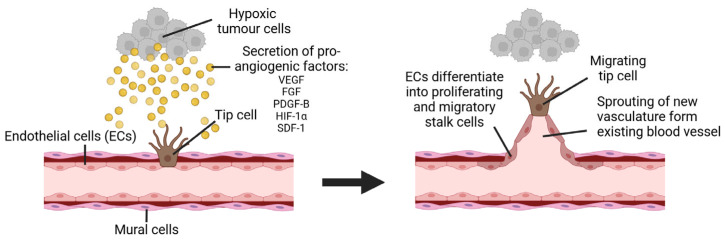
Tumorigenic neovascular development via angiogenesis process. Pro-angiogenic factors including VEGF, FGF, PDGF-B, HIF-1α, and SDF-1 are secreted by hypoxic cells in tumorigenic environments. These factors signal ECs to differentiate into a specialised type of EC known as tip cells, which guide the sprouting of new vasculature towards hypoxic environments. In turn, as tip cells migrate, ECs are differentiated into proliferating and migratory stalk cells that move along guiding tip cells to from newly sprouted vasculature from pre-existing blood vessels.

**Table 1 cancers-17-01675-t001:** Collection of miRNA cancer biomarkers (adapted from Cui et al. [83]).

Cancer Type	Expression Profile	Diagnostic Value	Prognostic Value	Diagnostic and Prognostic Value
Breast	Downregulated	let-7b-5p, let-7c-5p	miR-409-3p	
	Upregulated	miR-195, miR-376c, miR-409-3p, miR-148b, miR-299-5p, miR-145, miR-191, miR-382, miR-215, miR-133a, miR-133b, miR-92a, miR-192, miR-1, miR-411, miR-195, miR-202	miR-122 miR-141	miR-21, miR-34a, miR-210, miR-10b, miR-375, miR-125b, miR-801, miR-155
Pancreatic	Downregulated	miR-100-5p, miR-375	miR-718	
	Upregulated	miR-378 *, miR-409-3p, miR-1290, miR-26a, miR-18a	miR-146b-3p, miR-200a, miR-200c, miR-210, miR-221, miR-21, miR-194	miR-141, miR-375
Prostate	Downregulated	miR-16, miR-199a, miR-21		
	Upregulated	miR-378 *, miR-409-3p, miR-1290, miR-26a, miR-18a	miR-146b-3p, miR-210, miR-21, miR-221, miR-19, miR-200a, miR-200c	miR-141, miR-375
Non-small-cell lung carcinoma	Downregulated	let-7b-5p, let-7c-5p		
	Upregulated	miR-20a-5p, miR-141-3p, miR-145-5p, miR-155-5p, miR-223-3p	miR-320b, miR-23b-3p, miR-10b-3p, miR-195-5p	miR-21-5p

Note: Markers with (*) are also expressed in healthy populations but significantly upregulated in patients with specific cancer types.

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
