# Peer review of "Modelling Cancer Pathophysiology: Mechanisms and Changes in the Extracellular Matrix During Cancer Initiation and Early Tumour Growth"

_cancers, 2025, doi:10.3390/cancers17101675_

Round 1

Reviewer 1 Report

Comments and Suggestions for Authors

This review summarizes cancer transformation and angiogenesis very well, focusing on cancer-associated miRNAs and ECM remodeling, EMT, and EET. The role of the ECM in cancer development and early tumor growth is multifaceted, but I personally found Table 1 to be very informative, as it clearly summarizes previous reports on cancer-associated miRNAs, 2D vs. 3D culture in in vitro modeling, and microRNAs in tumor progression and ECM remodeling. The in vitro modeling of 2D or 3D cell cultures and the characteristics of microRNAs in tumor progression and ECM remodeling were important reports that might not have been summarized otherwise.

I have commented on a very minor point that caught my attention, and I would be fine if you could reconfirm it again. I hope that this review will reach young researchers of cancer/cancer therapeutics.

Comment:

  • (page 7, line246 and page 24 abbreviation) You wrote amino acid motif RGD as Gly-Arg-Gly-Asp-Ser, but I only knew Arg-Gly-Asp equal RGD. Are they needed in the sequence, the first Gly and last Ser? Please double check.

Author Response

Thank you for the feedback. We are very happy that you found our review informative and in particular the in vitro modelling of 2D or 3D cell cultures and the characteristics of microRNAs in tumour progression and ECM remodelling.

Yes, we agree with the reviewer, the RGD sequence equals Arg-Gly-Asp. This was a mistake on our part, and it has now been corrected on the main text (now Page 7, line 269), both when it was first introduced “Alginate functionalised with Arg-Gly-Asp (RGD), a short attachment...” and in the list of abbreviations, both highlighted in the manuscript.

Reviewer 2 Report

Comments and Suggestions for Authors

While the review offers a solid mechanistic understanding of the relationship between ECM, polarity, and miRNAs, the review could be further improved by looking at the metabolic-ECM interactions and the specific mechanisms that transform normal tissue into a cancerous lesion.

It would be helpful if the authors could address these key questions:

  1. Do cells in the center of an early tumour act differently from those at the invasive edge? (spatial heterogeneity)
  2. How do forces such as shear stress and fl uid pressure, in addition to matrix stiffness, affect the tumour?
  3. How do early tumors overcome the stiffness of the matrix to allow for invasion of cancer cells?

Author Response

Thank you for the feedback. We appreciate your comments regarding metabolic changes as well as other mechanotransduction signals associated with cancer development, maintenance and growth. We believe these are important topics to cover in our review and you have raised important questions we would like to address by adding the following paragraph between lines 136-160 which is highlighted in the revised manuscript:

Apart from matrix stiffness, mechanical forces such as shear stresses or fluid dynamics can significantly influence changes to the ECM and play a role in early tumour growth [37,38]. As solid tumours grow, they have been shown to accumulate mechanical stresses, in part due to proliferative cancer cells, but also due to radial and circumferential stresses from surrounding tissues. The accumulation of solid stresses can lead to the compression of intratumoral blood vessels and generate hypoxic cores within tumours. Such environments are frequently implicated in the induction of angiogenesis through hypoxia-inducible factor 1-alpha (HIF-1α) signalling, which facilitates the delivery of oxygen and/or nutrients to hypoxic cells, while concurrently hindering the effective distribution of chemotherapeutic agents due to compromised vascular perfusion. Both phenomena interlink to drive tumour progression [37,39]. The upregulated signalling of HIF-1α, induced by shear stresses within hypoxic environments, has been associated with the metabolic reprogramming of cells in tumorigenic environments, primarily by enhancing their glycolytic activity. This, in turn, promotes the growth, survival, proliferation, and long-term maintenance of tumorigenic environments [40,41]. However, mathematical models, such as those proposed by Mpekris et al., [39] have demonstrated that spatial heterogeneity exists within tumours.  Specifically, while the tumour core experiences an accumulation of compressive solid stresses, tensile forces are more prominent at the periphery. This heterogeneity in tissue mechanics was shown to correlate with increased vascular density and oxygenation at the tumour margins which in turn corresponded with higher proliferation rates of cancer cells compared to the core of tumours [39,42]. Furthermore, studies such as those by Lee at al., [43] and Hyler et al., [44] have shown that low levels of fluid shear stress can activate YAP/TAZ signalling pathways to drive cancer cell motility.  Additionally, such mechanical cues can induce structural and/or genomic alterations in benign cells, facilitating their transformation towards a cancerous phenotype. These findings highlight the importance of the mechanotransduction signals play in tumour development and progression.

References 37-44 have been added to support statements made in this paragraph which we hope will address the specific (1 and 2) questions raised in your comment. Although in this paragraph we do not specifically address your question about “how do early tumours overcome the stiffness of the matrix to allow for invasion of cancer cells”, we believe we address this in other sections of the review, specifically, where we mention the role MMPs, HIF and cancer-associated fibroblasts have in changing the ECM and how these changes facilitate the migration and invasion of cancer cells in various sections of the review.

Thank you again for your insightful comments and suggestions and we believe that the addition of the paragraph significantly improves the overall content of the manuscript.

Reviewer 3 Report

Comments and Suggestions for Authors

The submitted manuscript focuses on important issues related to the complex cancer initiation and development processes. However, improvements are needed to bring the manuscript into the form required by the scientific community.

The Simple Summary does not add any value, and its relatively long form, which is not inferior to the abstract, makes the reader doubt the real purpose of its presentation. It seems that the Simple Summary is a way of treating the reader as an immature recipient.

Figure 1 legend - "in red", "in blue", and "in green" should be in parentheses, especially as these markers are not the dominant markers in this figure.

Section 4 needs significant improvement. Many publications in the scientific literature describe the formation and role of miRNA in great detail, so it is necessary to remove general information and Figures 2 and 3, which contain information currently available in textbooks. The authors should focus only on issues strongly related to the manuscript topic.

Author Response

Thank you for the feedback and your thoughts that our manuscript focuses on  important issues related to the complex cancer initiation and development processes. We appreciate your comments regarding the Simple Summary, Figure Legend 1 and Section 4. We believe these comments are fair and the adjustments we made in these areas will improve our manuscript.

Specifically, the Simple Summary has been rewritten and cut down from 144 to 125 words to better reflect the readers’ previous understanding of cancer pathophysiology and related subjects:

Simple Summary: Cancer initiation and early tumour growth involve not only genetic mutations, but also important changes in the tumour microenvironment. Extracellular matrix (ECM) in particular undergoes biochemical and mechanical remodelling influencing cell polarity and abnormal vascular development to drive cancer initiation and tumorigenesis. MicroRNAs (miRNAs) further regulate these early processes by altering gene expression, promoting invasion, and contributing to tumour plasticity. Recent advances in in vitro modelling have aimed to replicate tumorigenic environments using interdisciplinary approaches to better recapitulate in vivo environments. This review discusses how ECM mechanics, cellular polarity loss, and miRNA dysregulation collectively contribute to tumour progression. It also highlights how current/emerging in vitro models, including self-assembling peptide hydrogels and bioprinting technologies, provide insight on strategies to recapitulate cancer initiation and early tumour growth.

Figure Legend 1 now reads “The polarity protein pathways during tissue homeostasis. The Par complex (red) localised to tight junctions, the scribble complex (blue) at the basal edge and the Crumbs complex (green) in the apical domain.” Parentheses have been added to “red”, “blue” and “green” in text and all authors agree that the legend reads better with those changes.

Section 4 has now been cut down to include important background and the role of miRNAs in cancer initiation and progression. In addition, Figure 2 has been removed, as per the reviewer’s suggestion. However, we have decided to retain Figure 3 (now Figure 2), as we believe it provides useful background context for readers. We feel that this figure supports the accessibility of the manuscript without detracting from its scientific focus.

We would like to thank you again for your comments. We think the comments were fair and hope the changes to the manuscript will reflect an improvement to the manuscript as well as address your concerns.